# The Role of G Protein-Coupled Estrogen Receptor (GPER) in Vascular Pathology and Physiology

**DOI:** 10.3390/biom13091410

**Published:** 2023-09-19

**Authors:** Fujie Xu, Jipeng Ma, Xiaowu Wang, Xiaoya Wang, Weiyi Fang, Jingwei Sun, Zilin Li, Jincheng Liu

**Affiliations:** 1Xi’an Medical University, Xi’an 710068, China; xufujie@fmmu.edu.cn (F.X.); fangweiyi@fmmu.edu.cn (W.F.); sunjingwei@fmmu.edu.cn (J.S.); 2Department of Cardiovascular Surgery, Xijing Hospital, Fourth Military Medical University, Xi’an 710032, China; majipeng@fmmu.edu.cn (J.M.); wangxiaowu@fmmu.edu.cn (X.W.); wangxiaoya@fmmu.edu.cn (X.W.)

**Keywords:** G protein-coupled estrogen receptor (GPER), estrogen, hypertension, atherosclerosis, inflammation, lipid metabolism

## Abstract

Objective: Estrogen is indispensable in health and disease and mainly functions through its receptors. The protection of the cardiovascular system by estrogen and its receptors has been recognized for decades. Numerous studies with a focus on estrogen and its receptor system have been conducted to elucidate the underlying mechanism. Although nuclear estrogen receptors, including estrogen receptor-α and estrogen receptor-β, have been shown to be classical receptors that mediate genomic effects, studies now show that GPER mainly mediates rapid signaling events as well as transcriptional regulation via binding to estrogen as a membrane receptor. With the discovery of selective synthetic ligands for GPER and the utilization of *GPER* knockout mice, significant progress has been made in understanding the function of GPER. In this review, the tissue and cellular localizations, endogenous and exogenous ligands, and signaling pathways of GPER are systematically summarized in diverse physiological and diseased conditions. This article further emphasizes the role of GPER in vascular pathology and physiology, focusing on the latest research progress and evidence of GPER as a promising therapeutic target in hypertension, pulmonary hypertension, and atherosclerosis. Thus, selective regulation of GPER by its agonists and antagonists have the potential to be used in clinical practice for treating such diseases.

## 1. Introduction

Cardiovascular disease (CVD) is currently the primary cause of death worldwide. In 2020, an estimated 19.05 million people died from CVD globally and 207.1 of 100,000 people died from hypertension and stroke, the prevalence of which was higher than that of cancer and chronic diseases [1]. Interestingly, epidemiological and clinical studies demonstrate clear sex differences for cardiovascular risk, prevalence, prognosis, pathophysiological manifestation, and response to treatment for CVD [2,3]. The incidence of cardiovascular disease is lower in premenopausal women than in age-matched men but sharply increases in postmenopausal women [4,5,6], implying that estrogen provides a protective effect and that the increased cardiovascular risk in women after menopause is related to the loss of the protective effect of estrogen [7,8]. The protection by estrogen was originally thought to be attributed to the classical estrogen receptors ER_α_ and ER_β_. However, the inhibition of these receptors hardly abolishes the protective effect of estrogen, indicating that a novel, unidentified receptor that was yet to be discovered was the actual mediator. This was later characterized as the G protein-coupled estrogen membrane receptor (GPER) [9,10].

GPER is present throughout the cardiovascular system, implying a critical role in maintaining cardiovascular physiological function. GPER activation by G1 restored glucose homeostasis and reduced body weight in multiple preclinical male and female models associated with obesity, diabetes, and metabolic abnormalities [11]. In blood vessels, GPER has important regulatory effects on the pathophysiological function of endothelial cells and smooth muscle cells [12,13]. Importantly, G1 treatment may regulate the renin–angiotensin system to reduce blood pressure in ovariectomized mRen2.Lewis rats [14]. These data collectively demonstrate that GPER plays a significant role in vascular-related diseases, including hypertension and atherosclerosis.

## 2. Tissue, Cellular Localization, Ligands, and Classical Signaling Pathways of GPER

GPER was initially identified as an orphan receptor with no known ligand, and belongs to the superfamily of 7-transmembrane G protein-coupled receptors (GPCRs). It was later named GPR30 based on the sequential numbering scheme for orphan receptors [15]. GPR30 was first shown to act as a functional estrogen receptor to mediate the fast activation of extracellular signal-regulated kinase (ERK) in the year 2000 [16]. Subsequently, experimental data from both groups showed that GPR30 directly bound to estrogen and activated its downstream intracellular signaling cascades [17,18]. Therefore, the International Union of Basic and Clinical Pharmacology officially named GPR30 as G protein-coupled estrogen receptor (GPER) [19].

### 2.1. Tissue and Cellular Localization of GPER

GPER is widely distributed in various tissues of the human body. GPER first attracted considerable attention in reproduction as an estrogen receptor that functioned in the breast [17,20], ovary [21], endometrium [22,23], testis [24], and prostate [25,26], as well as the placenta [27,28]. GPER is also found in other systems and organs, including the brain, lung, liver, heart, pancreatic islets, adipose tissue, vasculature, muscle, and skeleton, as well as in immune cells [29,30]. In the vascular system, GPER is distributed in both endothelial and smooth muscle cells of human arteries and veins [31,32,33].

The cellular localization of GPER has been a subject of much debate since its discovery. Most GPCRs are lipophilic, localized to the plasma membrane [34], and the signal acts by transferring from the outside to the inside of the cell across the plasma membrane. As a member of GPCRs, GPER was also initially thought to be located in the plasma membrane [35]. The immunohistochemical results obtained by Thomas et al. in breast cancer cells lacking ERα and ERβ were the first to confirm the plasma membrane localization of GPER [17]. Immunohistochemical analysis showed that GPER localized to the surface of CA2 cells [36]. Confocal microscopy using the agglutinin canavalin A as a marker of the plasma membrane also confirmed the expression of GPR30 on the cell surface [37]. Using RT-PCR and Western blot, it was found that GPER and calcin-1 were co-located on the cell membrane of human myometrium [38]. GPER was co-located with giant proteins on the membrane of proximal renal cortex tubules [39]. GPER was also detected in the dendritic spines of hippocampal neurons [40]. However, more studies have shown the intracellular localization of GPER. The intracellular localization of GPER was first discovered by Revankar et al. and was found to be localized in the inner membrane of the endoplasmic reticulum and Golgi apparatus of monkey kidney fibroblasts [18]. In the paraventricular and supraoptic nuclei, specific markers of intracellular organelles and immunoelectron microscopy revealed that GPR30 was mainly located in the Golgi apparatus of neurons, but could not be detected on the cell surface [41]. In transiently transfected cells, as well as in cells endogenously expressing GPR30, the GPER was shown to be localized to the endoplasmic reticulum [42]. In human umbilical vein endothelial cells (ECs) and vascular smooth muscle cells (VSMCs), GPER was shown to be expressed primarily on the endoplasmic reticulum and Golgi membrane [32]. Indirect evidence supporting the intracellular localization and functionality of GPER is that only cell-permeable E2 derivatives or intracellularly injected ligands resulted in rapid GPER activation [43], and intracellularly injected ligands produced faster and deeper calcium responses compared to extracellular ligands [44]. In some other cells, GPER was found expressing on the plasma membrane and inner membrane simultaneously. For example, almost all neurons in the trigeminal ganglion expressed GPER on the cell surface and in the cytoplasm [45]. Immunofluorescent results in human glioblastoma cells indicated that GPER was in the plasma membrane, cytoplasm, and nucleus [46]. In breast cancer cells, GPER has also been found to be localized in the nucleus, which is caused by the transport of GPER from the plasma membrane to the nucleus [47,48]. In addition, a prominent sex difference in the subcellular localization of GPER was detected in cardiomyocytes. In both wild-type and RenTgMK (a well-described model of angiotensin II-mediated chronic hypertension and cardiac hypertrophy [49,50,51]) female mice, the level of membrane localization of GPER protein was higher than in males of the same group [52]. Obviously, the localization of GPER is complex, which may be due to different species, sexes, and different tissues and cell types. It is worth noting that some studies have confirmed that GPER is activated within cells and then transported to the plasma membrane to play a role [53], which is regulated by co-expressed receptor activity-modifying protein 3 [52]. The transmembrane transport of GPER exerts physiological effects in a way that induces dynamic changes in the cellular localization of GPER, which undoubtedly makes the localization of GPER more complicated; thus, understanding the cellular localization of GPER is of importance for elucidating its function. 

### 2.2. Endogenous Ligands of GPER

#### 2.2.1. Estrogen

Estrogen is a steroid hormone which is secreted mainly from the ovaries, and four natural estrogens in females are biosynthesized: estrone (E1), 17β-estradiol (E2), 17α-estradiol (E2), and estriol (E3) [17,18,54]. Among them, 17β-estradiol is the most biologically active and common form of estrogen, and is primarily produced by the transformation of precursor testosterone in the ovaries of premenopausal women [15]. Estrogen is required for the development of female secondary sexual characteristics and affects the structure and function of the female reproductive system [55]. Estrogen was the first discovered ligand of GPER, and mediates pleiotropic functions by activating GPER in different tissues [56]. In the pathogenesis of diverse cardiovascular diseases, estrogen exerts important protective effects by activating GPER, which include the dilation of blood vessels, the prevention of atherosclerosis, a reduction in blood pressure, the suppression of myocardial hypertrophy, and the inhibition of myocardial ischemic injury [56]. Conversely, the absence of GPER exacerbates these pathologic reactions. The earliest experiments using *GPER* knockout mice showed that the loss of GPER increases blood pressure, eliminates E2-stimulated insulin release, and affects glucose metabolism, thereby leading to visceral obesity [57,58]. Hypertension due to GPER deficiency is associated with increased endothelial prostanoid-mediated vasoconstriction and increased contraction mediated by the thrombolin prostaglandin (TP) receptor [59]. In addition, *GPER* knockout mice showed an early increase in atherosclerotic progression [32]. In the heart, *GPER*-deficient mice exhibited impaired left ventricular cardiac function, left ventricular enlargement, and impaired systolic and diastolic function [60], or reduced ejection fraction and brachyaxis shortening, especially in older male mice [61]. Wang et al. showed that cardiomyocyte-specific *GPER* deletion caused left ventricular dysfunction and adverse remodeling in mice. Further specific spectral analysis showed that mitochondria-related genes were enriched in females of *GPER* KO mice, while genes related to the inflammatory response were enriched in male *GPER* KO mice [62]. This indicated that cardiac remodeling caused by GPER loss is associated with oxidative stress and NOD-like receptor thermal protein domain associated protein 3 (NLRP3)-mediated inflammatory activation [63,64]. GPER thus plays a protective function in the cardiovascular system.

GPER binds to estradiol with high affinity and was thought to mediate rapid non-genomic effects distinct from those of ER_α_ and ER_β_ [16,18,65,66]. Unlike most GPCRs, GPER couples to both a Gα_s_ protein and a pertussis toxin (PTX)-sensitive Gα_i/o_ protein, which activate intracellular signaling cascades using cAMP and Ca^2+^ as second messengers [16,17,65,66]. GPER is involved in the activation of multiple downstream cellular signaling cascades, as summarized in Figure 1. The first demonstrated signaling pathway activated by GPER involves the release of a tyrosine kinase (Src) to cleave matrix metalloproteinases (MMPs) which mediated heparin-binding epidermal growth factor (HB-EGF) activation. The subsequent transactivation of the epidermal growth factor receptor (EGFR) can further activate the ERK1/2 signaling pathway [67]. Through this mechanism, GPER mediates vasodilation by promoting NO production [68], induces smooth muscle cell proliferation [69], and promotes angiogenesis by upregulating angiotensin [70], and protects the myocardium from ischemia/reperfusion injury by reducing mitochondrial dysfunction and mitochondrial autophagy [71,72]. In addition, relevant studies in lung and breast cancer have shown that the activation of the Src/ERK1/2 pathway by GPER involves the activation of mitogen-activated protein kinase kinases (MEK) [73,74,75,76], which is consistent with our general understanding of the mitogen-activated protein kinase (MAPK) signaling pathway. Another classical pathway is phosphatidylinositol 3-kinase (PI3-K) and serine/threonine kinase (Akt), also known as protein kinase B (PKB), which appears upstream of ERK or as an independent signaling pathway and plays a role in cell proliferation [77], apoptosis [78,79], and cell migration [80,81]. It should be noted that ER_α_ also activates PI3K, which does not involve EGFR transactivation [18]. GPER was also found to activate the AMP-activated protein kinase (AMPK) signaling pathway, which enhances adenylyl cyclase (AC) activity to activate cAMP-activated protein kinase A (PKA). The PKA pathway has been shown to induce vasodilation in coronary vessels through GPER-mediated myosin light chain (MLC) phosphorylation and the ras homolog gene family, member A/ras homolog gene family (RhoA/Rho) kinase pathway [82,83]. Notably, the PKA pathway has been reported to have an inhibitory effect on ERK1/2 [65,84]. In addition, GPER has also been reported to inhibit inflammation [85,86,87], tumor cell migration, and tumor angiogenesis through reducing phosphorylation, nuclear localization, and the transcriptional activity of NF-κB, and TNF-α- or lipopolysaccharide-induced IL-6 secretion [88,89,90,91].

#### 2.2.2. Aldosterone

As a vasoactive steroid, aldosterone has recently attracted much interest, not only due to its great importance in the regulation of salt and water balance, but also because of its key role in cardiovascular metabolism and function. The classical effects of aldosterone are mediated by the activation of mineralocorticoid receptors (MRs) and subsequent transcriptional regulatory mechanisms [92]. However, aldosterone showed some rapid regulatory effects that cannot be explained by MRs [93], which appeared to involve a different receptor [94]. Gros et al. first described the role of GPER as a receptor involved in aldosterone-mediated signal pathways [95,96], indicating that GPER is an authentic aldosterone receptor.

Despite accumulating evidence for aldosterone’s actions through GPER, there remains significant speculation that aldosterone directly binds to GPER [97,98,99]. However, a recent work has demonstrated a direct interaction of aldosterone with GPER through the data showing that aldosterone competitively binds GPER with other GPER ligands [100]. Meanwhile, in a reperfusion injury study of the heart, using a polymer analogue of aldosterone to selectively activate the non-genomic receptors of aldosterone, it was found that the GPER antagonist G-36 but not spironolactone, a MR blocker, could block the action of aldosterone [101]. These data indicate that GPER, rather than MRs, was the receptor for the non-genomic action of aldosterone.

The mechanism of the downstream signal pathways via the interaction between aldosterone and GPER are not well understood. Recent studies have revealed that aldosterone trans-activates EGFR via GPER in adult rat ventricular myocytes, followed by reactive oxygen species (ROS) production and PI3K/AKT-dependent pathway activation, ultimately resulting in the stimulation of cardiac sodium/bicarbonate cotransporter (NBC) [102,103]. In addition, aldosterone also increases the cardiac vagal tone via GPER [104]. In the kidney, aldosterone has been reported to enhance the connecting tubule–glomerular feedback (CTGF) and stimulate oxygen production through the cAMP/PKA/PKC pathway in the connecting tubule (CNT) [105]. GPER in the distal tricyclic tubule (DCT) rapidly leads to intracellular cAMP and inositol phosphate accumulation, causing the activation of various downstream pathways including the MAPK/ERK, PI3K/AKT, and cAMP/PKA pathways, and the acute activation of thiazide-sensitive NaCl cotransporter (NCC) activity, thus regulating renal NaCl excretion [106]. 

GPER has also been shown to regulate aldosterone synthesis. Aldosterone increases the expression of *cyp11b2* mRNA through the activation of GPER, which is the last rate-limiting enzyme for aldosterone synthesis and up-regulates aldosterone secretion. This aldosterone-induced aldosterone production may, under certain circumstances, such as volumetric hypotension, physiologically increase aldosterone production to maintain blood pressure [107]. In contrast to GPER, ER_β_ was stated to inhibit aldosterone synthase expression and aldosterone synthesis through the finding that the selective blockade of ER_β_ markedly increased aldosterone expression, which was mimicked by the GPER-selective agonist G-1 and abolished by co-treatment with either the GPER antagonist G-15 or a selective protein kinase A inhibitor [108]. Thus, estrogen may exert different effects on aldosterone synthesis through different receptors, but exhibits inhibitory effects in physiological situations through ER_β_ [108]. In premenopausal women, high estrogen levels may maintain normotension by inhibiting aldosterone synthesis through ER_β_, and after menopause, the loss of estrogen can lead to an increase in blood pressure through GPER-mediated aldosterone release [109], which may suggest the cause of the high prevalence of hypertension in postmenopausal women. 

### 2.3. Exogenous Ligand of GPER

In addition to endogenous steroid hormones, many synthetic estrogenic compounds, as well as phytoestrogenic compounds of natural origin, have also been shown to bind GPER to exert their effects. Ligand-binding studies revealed that GPER binds E2 with much weaker binding affinities at concentrations from Kd = 3 to 6 nM [17] than classical ERs from Kd = 0.1 to 1.0 nM [110], so it was difficult to clarify the role of GPER alone in in vivo experiments without the synthetic highly selective GPER agonist G1 [111]. G-1 showed a stronger affinity (7–11 nM) and higher selectivity for GPER than E2 [111,112], and did not bind ER_α/β_ at concentrations up to 10 μM [113], which provides an extremely convenient tool to study the independent effects of GPER in vivo. To date, G-1 has been the most widely studied GPER agonist. A lot of the beneficial effects of GPER in the whole body are replicated by G-1, and unlike estrogen, G-1 can be considered a non-feminizing estrogenic compound due to its minimal effect on reproductive tissues; it thus has potential therapeutic use in both women and men [114]. Notably, the effect of G1 on GPER is concentration-dependent. G1 has other effects independent of GPER at micromolar concentration ranges. G1 (1–10 μM) inhibits the mitosis of human vascular smooth muscle cells and induces apoptosis by destroying microtubules [115]. Similarly, reduced DNA synthesis and microtubule structure disruption were observed in 2–3 μM G1-treated microvascular endothelial cells and HUVECs, which were non-receptor-dependent [116,117]. This concentration-dependent anti-proliferative effect was also observed in endometriosis stromal cells. G1 blocks microtubule assembly and induces cell cycle arrest in ESCs [118]. Therefore, in the application of G1 therapy or experimental treatment, attention should be paid to the concentration-dependent effect of G1 to prevent the influence of non-specific effects.

After the discovery of G-1, researchers subsequently synthesized the first GPER antagonist G-15 [112] and its modified analogue G-36 [113]. G-15 is a synthetic substituted dihydroquinoline with similar structure and a weak binding affinity for ER_α/β_ similar to G-1 [119] in the absence of the ethanone moiety [112]. Two additional compounds, GPER-L1 and GPER-L2, have been reported to act as novel GPER-selective agonists with a binding affinity of approximately 100nM [120]. The discovery of selective ligands for GPER has greatly helped us to understand the physiological role of GPER. In addition, numerous synthetic estrogenic compounds have been shown to bind and/or activate GPER, including bisphenol A [121,122,123,124], nonylphenol [124,125,126], atrazine [124,127,128], kepone, methoxychlor, p,p’-DDT, o,p’-DDT, o,p’-DDE, p,p’-DDE, 2,2’5’-PCB-4-OH, etc. [17,124]. Lastly, there are some plant-derived polyphenols that have also been shown to bind to GPER, including Genistein [124,129,130,131], Daidzein [132], Equol [133], Oleuropein [134], Hydroxytyrosol [133], Resveratrol [135,136,137], Quercetin [136,138], Tectoridin [139], Zearalenone [124,140,141], and Apigenin [136,142]. The structure and function of GPER ligands are summarized in Table 1.

## 3. The Role of GPER in Vascular Pathology and Physiology

### 3.1. Blood Pressure Regulation and Hypertensive Disorders

Hypertension (HTN) is a multifactorial disease, defined as a systolic blood pressure (SBP) greater than 130 mmHg or diastolic blood pressure (DBP) higher than 80 mmHg [146,147], and is an important risk factor for cardiovascular diseases such as coronary heart disease, heart failure, and stroke. HTN is characterized by persistent elevated blood pressure levels and increased peripheral vascular resistance [148,149]. The gene of GPER is mapped to chromosome 7p22 [150], a region associated with human arterial hypertension [151], which suggests a correlation between GPER and blood pressure control. Data have demonstrated that GPER is associated with hypertension in postmenopausal women [152]. The difference in gender hypertension may be related to the impaired function of GPER, since the frequency of the P16L allele, a lower-functioning variant of GPER, is statistically almost twice as high in female hypertension patients as in male patients [153]. However, aldosterone-induced hypertension has sex differences, which is mediated by T cells and attenuated by the non-T cell GPER-dependent pathway [154]. Aortic remodeling caused by salt-sensitive hypertension was also ameliorated via GPER activation [155].

Animal experiments proved that GPER could mediate vasodilation in rats and reduce blood pressure in normotensive rats [58]. In the hypertensive rat model, G1 treatment was able to reduce blood pressure by dilating rat mesenteric arteries [156]. This rapid blood-pressure-lowering effect of GPER protects the heart from hypertensive damage [58]. The estrogen-induced endothelium-dependent vasodilation was revealed to correlate with GPER-mediated nitric oxide (NO) production [13]. As a mediator of acute estrogen-dependent vasodilation, GPER is responsible for the control of vascular tone involving NO and hypotensive activity [157]. It is an important regulator of endothelial cells to regulate the tension of resistance vessels [158]. GPER activation induces endothelial cells to produce NO, which mediates vasodilation in a variety of blood vessels, including carotid artery rings of Sprague Dawley rats [58], isolated aortic and mesenteric rings of mRen2 mice [39,156], aortic rings [96,159,160] and mesenteric arteries of rats [66], and rat uterine arteries [161,162]. Both estrogen and aldosterone can activate GPER to exert a rapid vasodilator effect [160,163], but the vasodilator effect of aldosterone is almost entirely dependent on GPER [96]. Notably, the estrogen-mediated activation of eNOS is not only dependent on GPER, but also dependent on ER_α_ [164], but GPER is more involved in estrogen-induced endothelial vasodilation and NO production [13]. 

NO is produced in endothelial cells, which involves the phosphorylation of eNOS caused by Src/EGFR/PI3K–Akt and ERK pathways [13,68]. Pollyana et al. showed that the GPER-dependent relaxation of mesenteric resistance arteries is mainly mediated by the PI3K–Akt–eNOS pathway and attenuated by non-specific potassium channel block [165]. The molecular mechanism of GPER-dependent activation of the ERK1/2, PI3K-Akt, and NO signaling pathways was also supported by dibutyl phthalate (DBP)-induced endothelial cell proliferation [70]. Notably, there appear to be sex differences in the mechanistic pathways involved in eNOS phosphorylation: the PI3k–Akt–eNOS pathway was identified in males, and the MEK–ERK–eNOS pathway was validated in females [166]. In addition to this, NO production can also occur through genomic pathways. The underlying mechanisms include the activation of GPER which increases intracellular Ca^2+^ levels through transient receptor potential channel (TRPC) isoforms, leading to the phosphorylation of calmodulin-dependent protein kinase 2 (CaMKKβ), AMPK, and calmodulin-dependent protein kinase II (CaMKIIα), which in turn increase the expression of eNOS through the activation of histone deacetylase 5 (HDAC5) and the Kruppel-like factor 2 (KLF2) signaling pathway [68]. GPER also increased the phosphorylation of EGFR by interacting with Src, which activates eNOS through ERK5 and muscle cell enhancer 2 peptide C (MEF2C) [68]. After endothelium-dependent NO production, VSMCs are stimulated, causing downstream guanylate cyclase-mediated cGMP release and protein kinase G stimulation, followed by Ca^2+^ influx channel opening and myosin light chain phosphatase (MLCP) activation [66,167] (Figure 2). Yu et al. revealed that G-1 exerts a concentration-dependent relaxation effect on isolated porcine coronary arteries by increasing potassium efflux from the large-conductance calcium-activated potassium (BKCa) channel, which was not endothelium-dependent [168]. Another report showed that the activation of GPER inhibited the RhoA/Rho kinase pathway through the cAMP/PKA signaling pathway, which in turn activated MLCP to mediate vasodilation in isolated porcine coronary arteries [82]. In addition, they identified an additional GPER-dependent exchange protein directly activated by camp 1/ras-related protein 1 (Epac/Rap1) pathway parallel to PKA, which inhibited the RhoA/Rho kinase pathway [83]. Later, they demonstrated that cAMP production was dependent on EGFR transactivation and ERK1/2 phosphorylation, which mediated the relaxation of isolated porcine coronary arteries [169]. In conclusion, GPER mediates vascular relaxation through RhoA/Rho inhibition, MLCP activation, and Ca^2+^ channel opening via the endothelium-dependent PKG pathway and the endothelium-independent PKA and Epac/Rap1 pathways (Figure 2). Indeed, the synergistic effect of PKA and PKG on VSMC relaxation is evidenced by vasodilation being inhibited by the administration of cAMP and cGMP antagonists [170] and vasodilation being partially abolished in mesenteric arteries by the administration of AC and GC inhibitors [66]. Due to the vasodilation and protective effects of GPER on blood vessels, GPER and its related pathways would be potential therapeutic targets for the treatment of hypertension, especially in postmenopausal women. 

### 3.2. Pulmonary Arterial Hypertension

Pulmonary arterial hypertension (PAH), defined as elevated mean pulmonary arterial pressure (PAP, typically >25 mmHg at rest, or >30 mmHg with exercise, or >40 mmHg for systolic), is characterized by the pathological elevation of pulmonary artery pressure leading to remodeling of the right ventricle and ultimately to severe right heart failure [171]. The prevalence of PAH is higher in women [172], thus implying the vital role of estrogen in the pathogenesis of PAH. However, a large amount of experimental evidence has shown that women have a lower severity of PAH than men, as well as a better prognosis. The symptoms of PAH are aggravated after ovariectomy, indicating that estrogen has an inhibitory effect on the occurrence and development of PAH. This phenomenon is called the “estrogen paradox” [173,174,175]. The protective effect of estrogen on pulmonary hypertension has been demonstrated [174,175], but as a new membrane receptor of estrogen, the effect of GPER on PAH is still under investigation. Interestingly, the activation of GPER by G1 demonstrated beneficial effects on PAH-related cardiopulmonary functional and structural abnormalities in both genders. In male rats, it was mediated by improving pulmonary endothelial nitric oxide synthesis, Ca^2+^ handling regulation, a decrease in inflammation in cardiomyocytes, and a decrease in collagen deposition by acting in pulmonary and cardiac fibroblasts [176]. In female rats, it was mediated by reducing right ventricular collagen deposition and hypertrophy via decreasing glycosaminoglycan content and reactive oxygen species production in female rats [177].

### 3.3. Inflammation, Lipid Metabolism, and Atherosclerosis

Atherosclerosis is characterized by plaque formation leading to the stenosis of the arterial wall, and is associated with a long-term state of dyslipidemia and vascular inflammation [178]. Correlations between GPER and inflammation, lipid metabolism, and atherogenesis have been demonstrated [32]. A study using B-ultrasound to investigate atherosclerosis in postmenopausal women showed a significantly lower prevalence of atherosclerosis in late menopausal women and in women receiving postmenopausal estrogen therapy compared to early menopausal women and women without postmenopausal estrogen therapy [179]. The absence of GPER in male mice may generate some pathological changes, such as insulin resistance, dyslipidemia, and inflammatory effects [180]. Decreased high-density lipoprotein (HDL) cholesterol and increased triglyceride levels were observed in *GPER*-deficient female mice, and aged male *GPER*-deficient mice exhibited dyslipidemia and inflammation [180]. In a pathological model of atherosclerosis in female mice, a loss of GPER increased total cholesterol (TC) and low-density lipoprotein (LDL) levels [32], and treatment with G-1 was effective in attenuating atherosclerosis in postmenopausal mice, which was associated with a significant reduction in vascular inflammation [32]. The effects of GPER on inflammation, lipid metabolism, and atherosclerosis have not only been validated in animal models but also in humans. In a cohort study, humans carrying a low-functioning P16L genetic variant of GPER were found to have increased plasma LDL cholesterol [181]. Meanwhile, in the human hepatic carcinoma cell line, G1 treatment led to a concentration-dependent elevation in LDL receptor expression, and the effects of G1 were attenuated by G15 as a GPER antagonist or GPER shRNA [181], suggesting that GPER plays an important role in regulating LDL receptor expression and LDL metabolism. Ghaffari et al. used total internal reflection fluorescence microscopy to quantify the transcellular transport of LDL in human coronary-artery endothelial cells obtained from multiple donors. The rate of transcytosis was higher in coronary artery endothelial cells isolated from male patients compared to premenopausal female patients [182]. Estrogen administration reduced LDL transcytosis in endothelial cells from male but not female donors, and the inhibition of ER_α_ and ER_β_ had no effect on the estrogen-mediated attenuation of LDL trans-endocytosis. Instead, estrogen’s effect on LDL transcytosis was blocked by the depletion of GPER [182]. Apparently, the effects of estrogen on LDL are mediated through GPER but not ER_α_ or ER_β_.

Further evidence for the anti-inflammatory effects of GPER in blood vessels and inflammatory cells was obtained in in vitro studies. In human umbilical vein vessels, GPER activation attenuated the TNF-α-induced upregulation of proinflammatory proteins, which was reversed by G-15 [183]. However, in an estrogen-deficient environment, GPER-mediated changes in the TRPC1/ERK1/2 pathway may cause an increased inflammatory response, which is associated with the development of atherosclerosis in postmenopausal women [184]. In cultured rat aortic smooth muscle cells, the up-regulation of GPER expression and down-regulation of ER_α_ both reduced medial vascular hypertrophy and inflammatory processes after carotid artery ligation [185]. The selective GPR30 agonist G-1 suppressed the TNF-induced upregulation of pro-inflammatory proteins, which was completely abrogated by the selective GPR30 antagonist G-15 [183]. Remarkably, estrogen treatment had no effect on the TNF-treated endothelium, suggesting that the activation of the classical ER may block the anti-inflammatory effects of GPER [183]. The selective ligands of GPER reduced the C-reactive protein (CRP)-mediated expression of inflammatory mediators in primary cultured bone-marrow-derived macrophages (BMM) and VSMCs [186]. In addition, the anti-inflammatory effects of GPER have been confirmed by several systemic disease models. In multiple sclerosis models, a loss of GPER exacerbates the inflammatory response, which is reduced by treatment with G1 [187]. In autoimmune encephalomyelitis, estradiol treatment reduced the inflammatory response in *ER_α_* knockout mice but not in *GPER* knockout mice [188]. In a hepatocyte mitochondrial injury model, GPER activated the AMPK–Nrf2 signaling pathway to alleviate the oxidative stress and the inflammatory response [85]. Indeed, the immunomodulatory function of GPER would be expected for the reason that GPER was cloned from a human B cell lymphoblast cell line cDNA library [15], and is expressed and functions in leukocytes [189] and macrophages [190]. 

It is well accepted that the maintenance of the integrity and physiological function of endothelial cells is vital for vascular function, the dysregulation of which is the initial step for the development of atherosclerosis [191]. In a rabbit model of atherosclerosis induced by a high-fat diet and compression balloon, GPER reduced arterial lipid levels including total cholesterol (TC), low-density lipoprotein (LDL), and triglycerides (TG), and increased high-density lipoprotein (HDL) through inhibiting endothelial apoptosis and endothelial dysfunction via the PI3K/Akt pathway [192]. In ovarian *ApoE*-deficient mice, GPER attenuated endothelial cell apoptosis, which reduced plasma lipid levels and prevented atherosclerotic plaque formation [193]. The kaempferol-induced upregulation of GPER attenuated atherosclerosis by inhibiting inflammation and apoptosis through the PI3K/AKT/Nrf2 pathway [194]. The inhibition of the proliferation and migration of VSMCs is one of the key events in the prevention of early atherosclerosis [195,196]. G-1 reduced serum-stimulated proliferation of human vascular smooth muscle cells, which was abrogated by *GPER* deletion [58]. In cultured rat aortic smooth muscle cells, both estradiol and G1 inhibited the cell proliferation and migration, which was blocked by the GPER antagonist G15 [185]. Estrogen prevented the apoptosis of VSMCs, thereby inhibiting the proliferation of human cerebral VSMCs by up-regulating GPER and ERs and down-regulating caspase-3, myocardin (MYOCD), and serum reaction factor (SRF) [197]. In addition, the activation of GPER was shown to inhibit vascular endothelial growth factor (VEGF) and tumor angiogenesis in triple-negative breast cancer (TNBC) [198]. Through inhibiting VSMC proliferation and migration, as well as its anti-apoptosis effects, GPER contributes to the maintenance of vascular homeostasis. Given that the activation of GPER improves lipid metabolism, regulates endothelial and vascular smooth muscle cell function, and suppresses inflammation, it is a promising therapeutic target for the treatment of atherosclerosis. The research on the role of GPER in the vascular pathology is summarized in Table 2.

## 4. Conclusions and Future Perspectives

As an estrogen receptor, GPER is widely distributed throughout the body, and its beneficial effects have been demonstrated in cardiovascular diseases. The binding of GPER to aldosterone has been demonstrated, but the exact mechanism remains to be elucidated. In addition, many natural and synthetic molecules are also able to activate or inhibit GPER, but further studies are needed to determine how they affect disease and health through GPER. In blood vessels, GPER is indispensable in the physiological functions of endothelial cells and smooth muscle cells. Given that the incidence of CVD is lower in premenopausal women than in age-matched men, it is likely that estrogen plays a crucial role in this disparity [6]. However, due to the complexity of the physiological action of estrogen itself, estrogen imbalance can cause many diseases [199,200]. Moreover, due to the complexity of estrogen receptors, the same estrogen acting on different estrogen receptors can have different effects, some of which are beneficial and others of which are harmful [201]. Clinically, hormone replacement therapy (HRT) is a common treatment option [202], but clinical studies have shown that postmenopausal estrogen therapy increases the risk of stroke [203,204], myocardial infarction, and deep vein thrombosis (DVT) and/or pulmonary embolism (PE) [205]. Therefore, highly selective estrogen receptor agonists and antagonists are promising candidates for drugs. Furthermore, since selective GPER agonists, unlike natural estrogens [206], do not have uterine trophic effects [112,207], postmenopausal women may also receive GPER-targeted therapy to provide vascular protection. 

## Figures and Tables

**Figure 1 biomolecules-13-01410-f001:**
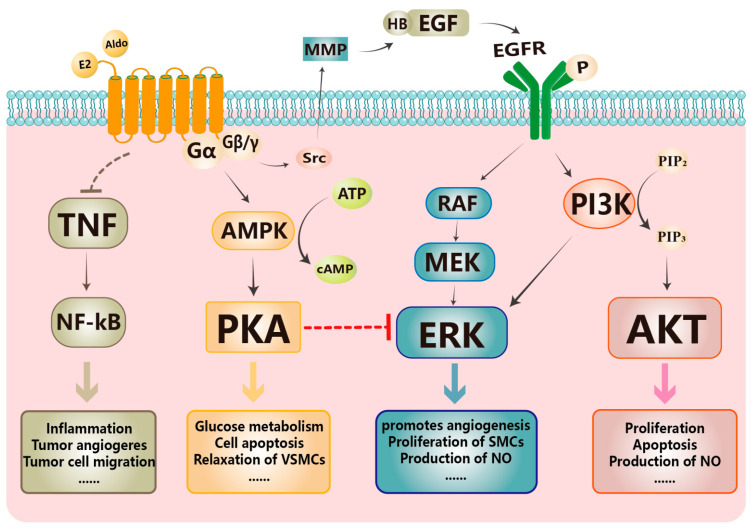
Classical signaling pathways activated by GPER. GPER binds to its endogenous ligands (estrogen and aldosterone), leading to multiple downstream cascades, including ERK, PI3K/AKT, AMPK/PKA, and NF-kB pathways. Activation of Src via GPER results in the release of matrix metalloproteinase (MMP) mediation of heparin-binding epidermal growth factor (HB-EGF) cleavage, followed by transactivation of the EGF receptor (EGFR), which in turn causes the activation of extracellular signal-regulated kinases (ERK) and the activation of phosphatidylinositol 3-kinase (PI3K) and serine/threonine kinase (PI3K/AKT). GPER stimulates the AMPK signaling pathway, which increases adenylyl cyclase (AC) activity to stimulate the cAMP-activated downstream protein kinase A (PKA) signal pathway that inhibits ERK. GPER also reduces the phosphorylation, nuclear localization, and transcriptional activity of the NF-κB pathway.

**Figure 2 biomolecules-13-01410-f002:**
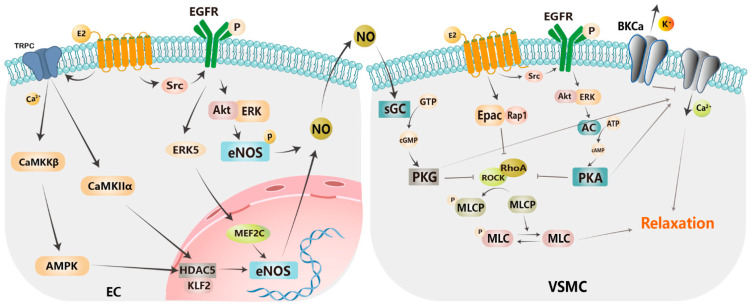
GPER-mediated relaxation of VSMCs. Activation of GPER increases intracellular Ca^2+^ levels through transient receptor potential channel (TRPC) isoforms, leading to the phosphorylation of calmodulin-dependent protein kinase 2 (CaMKKβ), AMP-activated protein kinase (AMPK), and calmodulin-dependent protein kinase II (CaMKIIα), which in turn elevates the expression of eNOS through the activation of histone deacetylase 5 (HDAC5) and the Kruppel-like factor 2 (KLF2) signaling pathway. GPER also enhances the phosphorylation of EGFR by interacting with various proteins around the membrane, such as Src, which further induces the expression of eNOS through ERK5 and muscle cell enhancer 2 peptide C (MEF2C). GPER phosphorylates eNOS through the non-genomic Akt/ERK pathway, which in turn promotes NO production. NO produced in ECs stimulates sGC in VSMCs to produce cGMP, which in turn activates PKG. PKG cooperates with Akt/ERK-dependent PKA and Epac/Rap1 pathways to inhibit RhoA/Rho, thereby causing MLCP activation and Ca^2+^ channel opening, ultimately leading to VSMC relaxation. EC, endothelial cell; VSMC, vascular smooth muscle cell.

**Table 1 biomolecules-13-01410-t001:** The structure and function of GPER ligands.

Compound	Structure	Agonist/ Antagonist	Affinity/Efficacy K_d_/IC_50_/EC_50_	Reference(s)
Endogenous Ligand				
Estrogen				
Estrone (E1)	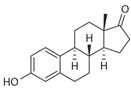	Unknown	>>10 μM	[17]
Estradiol (E2)				
17β-Estradiol	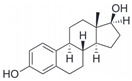	Agonist	3–6 nM	[17,18]
17α-Estradiol	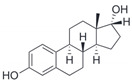	Unknown	>10 μM	[17]
2-Methoxy-E2	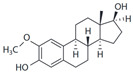	Agonist	10 nM	[143]
2-Hydroxy-E2	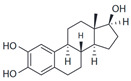	Antagonist	0.1–1 μM	[144]
Estriol (E3)	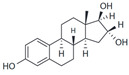	Antagonist	>1 μM	[17,54]
Aldosterone	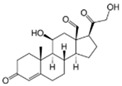	Agonist	None detected	[95,104,145]
Synthetic Ligands				
G1	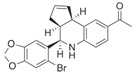	Agonist	7–11 nM	[111,112]
G15	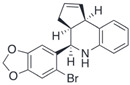	Antagonist	20 nM	[112,119]
G36	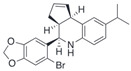	Antagonist	~ 20 nM	[113]
GPER-L1	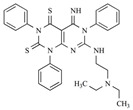	Agonist	100 nM	[120]
GPER-L2	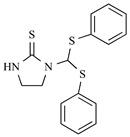	Agonist	100 nM	[120]
Bisphenol A	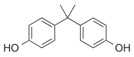	Agonist	0.6 μM	[121,122,123,124]
Nonylphenol	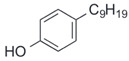	Agonist	0.8 μM	[124,125,126]
Atrazine	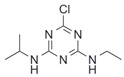	Agonist	>10 μM	[124,127,128]
Kepone	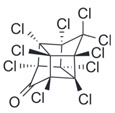	Agonist	1.4 μM	[124]
Methoxychlor	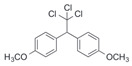	Unknown	~10 μM	[124]
p,p′-DDT	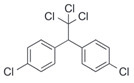	Unknown	2.8 μM	[124]
o,p′-DDT	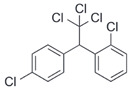	Agonist	7.1 μM	[17,124]
p,p′-DDE	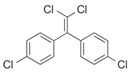	Unknown	~10 μM	[124]
2,2′5′-PCB-4-OH	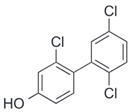	Unknown	3.8 μM	[124]
Phytoestrogens				
Genistein	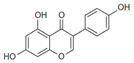	Agonist	133 nM	[124,129,130,131]
Daidzein	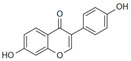	Agonist	<1 nM	[132]
Equol	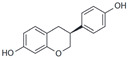	Agonist	100 nM	[133]
Oleuropein	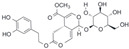	Agonist	200 nM	[134]
Hydroxytyrosol	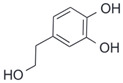	Agonist	100 nM	[133]
Resveratrol	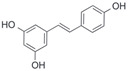	Agonist	300 nM	[135,136,137]
Quercetin	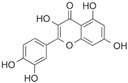	Agonist	1 μM	[136,138]
Tectoridin	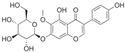	Agonist	10 μM	[139]
Apigenin	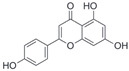	Agonist	20–50 μM	[136,142]
Zearalenone	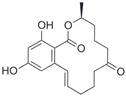	Unknown	0.8 μM	[124,140,141]

**Table 2 biomolecules-13-01410-t002:** The role of GPER in vascular pathology.

Model	Intervention	Outcome	Reference
Hypertension			
Human endothelial cells	GPER agonist	GPER↑→eNOS↑→Src/EGFR/PI3K/ERK↑→NO↑→Vasodilation↑→HTN↓	[13]
Human endothelial cells	GPER antagonist	GPER↓→NO↓→Vasodilation↓→HTN↑	[13]
Human endothelial *EA.hy2* cells	GPER agonist	GPER↑→CaMKKβ, AMPK, CaMKIIα phosphorylation↑→HDAC5, KLF2 →eNOS↑→NO↑→Vasodilation↑	[68]
Pig coronary artery	GPER agonist	GPER↑→cAMP/PKA↑→RhoA↓→MLCP↑→Vasodilation↑	[82]
Pig coronary artery	GPER agonist	GPER↑→Epac/Rap1↑→RhoA/Rho↓→Vasodilation↑	[83]
Rat aortic smooth muscle cells	*GPER* mutation	ERK↓, Apoptosis↓→ HTN↑	[153]
Aldosterone-induced hypertension in mice	*GPER* KO or antagonist	GPER↓→HTN↑	[154]
*mRen2* female hypertensive rat	GPER agonist	GPER↑→Oxidative stress↓→Aortic remodeling↓	[155]
Mesenteric artery of rat	GPER agonist	GPER↑→NO↑, H_2_O_2_↑→Vasodilation↑	[156]
Mesenteric artery of rat	GPER agonist	GPER↑→PI3K/Akt/eNOS↑→NO↑→Vasodilation↑	[165]
Mesenteric artery of male rat	GPER agonist	GPER↑→PI3K/Akt/eNOS↑→NO↑→Vasodilation↑	[166]
Mesenteric artery of female rat	GPER agonist	GPER↑→MEK/ERK/eNOS↑→NO↑→Vasodilation↑	[166]
Human or pig coronary smooth muscle cells	GPER agonist	GPER↑→BK (Ca)↑→Potassium efflux↑→Vasodilation↑	[168]
Pulmonary arterial hypertension			
Male rats with MCT-induced PH	GPER agonist	GPER↑→eNOS↑, Collagen deposition in pulmonary and cardiac fibroblasts↓, Ca^2+^↑, Inflammation in cardiomyocytes↓→Pulmonary flow↑, RV hypertrophy↑, LV dysfunction↑	[176]
OVX female rats with MCT-induced PH	GPER agonist	GPER↑→Pulmonary artery dysfunction↓, RV overload↓, RV dilation↓, Wall hypertrophy↓, Collagen deposition↓, Normalizes LV dysfunction	[177]
Atherosclerosis			
Mice fed high-fat diet	*GPER* KO or OVX	GPER↓→TC↑, LDL↑, Inflammation↑, NO bioactivity↓→AS↑	[32]
Mice fed high-fat diet	GPER agonist	GPER↑→Inflammation↓→AS↓	[32]
Male and female mice	*GPER* deletion	GPER↓→ endothelium-dependent vasoconstriction↑, visceral obesity↑, LDL↑, inflammation↑	[67]
Male mice	*GPER* KO	GPER KO→Insulin resistance, Dyslipidemia, Inflammatory effects↑	[180]
Female mice	*GPER* KO	GPER KO→HDL↓, TG↑	[180]
Human coronary artery ECs	*GPER* knockdown	GPER↑→EGFR↑→endothelial scavenger receptor class B type I↓→LDL transcytosis↓	[182]
Female mice	GPER agonist	Ca↑→GPER↓→TRPC1/ERK1/2↓→AS↑	[184]
Rat aortic SMCs	GPER agonist, *ERα* knockdown	Hypertrophy and inflammation of medial vessels after carotid ligation↓	[185]
Rabbit atherosclerotic ECs	GPER agonist	GPER↑→PI3K/Akt↑→Inflammation↓, Apoptosis↓→Dysfunction of ECs↓→TC↓, TG↓, LDL↓, HDL↑→AS↓	[192]
*ApoE* mice	GPER agonist	GPER↑→Apoptosis of ECs↓→Plasma lipid↓→AS↓	[193]
*ApoE* mice	GPER agonist	GPER↑→PI3K/Akt/Nrf2↑→Inflammation↓, Apoptosis↓→AS↓	[194]

GPER: G protein-coupled estrogen receptor; eNOS: endothelial nitric oxide synthase; Src: tyrosine kinase; EGFR: epidermal growth factor receptor; PI3K: phosphatidylinositol 3-kinase; ERK: extracellular signal-regulated kinase; NO: nitric oxide; HTN: hypertension; CaMKKβ: calmodulin-dependent protein kinase 2; AMPK: AMP-activated protein kinases; CaMKIIα: calmodulin-dependent protein kinase II; HDAC5: histone deacetylase 5; KLF2: Kruppel-like factor 2; cAMP: cyclic adenosine monophosphate; PKA: protein kinase A; MLCP: myosin light chain phosphatase; Epac/Rap1: exchange protein directly activated by camp 1/ras-related protein 1; RhoA/Rho: ras homolog gene family, member A/ras homolog gene family; KO: knockout; H_2_O_2_: hydrogen peroxide; Akt: serine/threonine kinase; MEK: mitogen-activated protein kinase kinase; BK (Ca): large-conductance calcium-activated potassium; RV: right ventricle; LV: left ventricle; OVX: ovariectomy; MCT: monocrotaline; TC: total cholesterol; ECs: endothelial cells; LDL: low-density lipoprotein; AS: atherosclerosis; HDL: high-density lipoprotein; TG: triglyceride; TRPC1: transient receptor potential channel 1; SMCs: smooth muscle cells; Nrf2: nuclear factor-erythroid 2-related factor 2; ApoE: apolipoprotein E-deficient; ↑: increased; ↓: decreased; →: results in.

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
