# Peer review of "The Role of G Protein-Coupled Estrogen Receptor (GPER) in Vascular Pathology and Physiology"

_biomolecules, 2023, doi:10.3390/biom13091410_

Round 1

Reviewer 1 Report

A very in-depth review which systematically explores the role of GPER within the cardiovascular system and in pathophysiologies associated with CVD. The review is well written and informative. It would provide readers with good insights about GPER role in regulating cardiovascular -function and -disease. I have a few suggestions that the authors should consider adding and expanding upon.

1: The authors should provide some data from GPER knockout models. Are there deleterious effects on the CVD ? what is known ? does it support the role of GPER as a protective E2 receptor.

2: As pointed by the authors, G1 has been extensively used to study the role of GPER in the cardiovascular system. Although G1 has specificity for GPER and does not engage ER alpha or Beta, it does have other GPER independent actions at concentrations higher than nmolar range. It seems to interact or interfere with tubulin and actin dynamics and influence cell growth. Hence one must point out that the effects of G1 at higher that nM range may be non-specific and participation of other mechanism cannot be ruled out. For example in growth studies G1 has no effect on SMC growth within nM range, however the growth inhibitory effects show up at micromolar concentrations that interfere with tubulin. Hence data form G1 has to be taken with a grain of salt as studies have taken its specificity for granted and never looked at the alternative mechanism. The authors should add a section on limitations and what needs to be further proven beyond doubt and add something about the non-specifc actions that may play a role too.      

none

Reviewer 2 Report

It is worth pointing out in the abstract what the review shows.

Some sentences should be made shorter. Example:

This may be due to differences in the cell 74 types and methods. Although the results from several studies are consistent with the con- 75 clusion of plasma membrane localization[38, 39], and even nuclear localization has been 76 observed [40, 41], most studies demonstrate that GPER is localized on the endoplasmic 77 reticulum membrane [40, 42-45].

It is necessary to indicate in more detail what these references indicate.

Many sentences are too general. Describe the evidence you indicate. Example:

“However, numerous evidences suggest that GPER-activated signaling cascades can ulti- 119 mately regulate gene expression via transcriptional mechanism [66, 76-79]”

Research on the role of GPER in pathology would be worth placing in one summary table.

Minor editing of English language required
